# Extracellular Vesicles Mediate B Cell Immune Response and Are a Potential Target for Cancer Therapy

**DOI:** 10.3390/cells9061518

**Published:** 2020-06-22

**Authors:** Taketo Kato, Johannes F. Fahrmann, Samir M. Hanash, Jody Vykoukal

**Affiliations:** 1Department of Clinical Cancer Prevention, The University of Texas MD Anderson Cancer Center, 1515 Holcombe Boulevard, Houston, TX 77030, USA; tkato1@mdanderson.org (T.K.); jffahrmann@mdanderson.org (J.F.F.); shanash@mdanderson.org (S.M.H.); 2McCombs Institute for the Early Detection and Treatment of Cancer, The University of Texas MD Anderson Cancer Center, Houston, TX 77030, USA

**Keywords:** extracellular vesicles, tumor-associated antigens, autoantibody, B cell response, cancer

## Abstract

Extracellular vesicles (EVs) are increasingly understood to participate directly in many essential aspects of host antitumor immune response. Tumor- and immune-cell-derived EVs function in local and systemic contexts with roles in immune processes including cancer antigen conveyance, immune cell priming and activation, as well as immune escape. Current practice of cancer immunotherapy has de facto focused on eliciting T-cell-mediated cytotoxic responses. Humoral immunity is also known to exert antitumor effects, and B cells have been demonstrated to have functions that extend beyond antibody production to include antigen presentation and activation and modulation of T cells and innate immune effectors. Evidence of B cell response against tumor-associated antigens (TAAs) is observed in early stages of tumorigenesis and in most solid tumor types. It is known that EVs convey diverse TAAs, express antigenic-peptide-loaded MHCs, and complex with circulating plasma antitumoral autoantibodies. In this review, we will consider the relationships between EVs, B cells, and other antigen-presenting cells, especially in relation to TAAs. Understanding the intersection of EVs and the cancer immunome will enable opportunities for developing tumor antigen targets, antitumor vaccines and harnessing the full potential of multiple immune system components for next-generation cancer immunotherapies.

## 1. Introduction

Converging lines of evidence reveal extracellular vesicles (EVs) as important mediators of tumor-immune interchange [1,2,3,4,5]. Profiling tumor-derived EVs (TDEs) reveals a diverse repertoire of tumor-associated antigens (TAAs) and functional effectors of humoral and cellular immunity [6]. The findings suggest EVs as a compelling target for antitumor intervention, both as a means to guide existing modes of immunotherapy through interrogation of tumor surface antigens or status of immune infiltrate, or as a novel target for interception and reversal of cancer cell signaling in relation to immune tolerance.

Engaging host immune system function to effect antitumor immunity is an important emergent axis of cancer therapy. Efforts in this regard have concentrated primarily on augmenting T cell responses either through cytokine activation of T cells, inhibition of immune checkpoint mediators that attenuate T cell cytotoxic function, or through ex vivo expansion and reintroduction of tumor resident or genetically modified tumor-targeting T cells [7]. Although these approaches have led to unprecedented positive outcomes in many cases, clinical data indicate that a majority of patients do not benefit from these treatment modalities, thereby accentuating the need to develop additional classes of cancer immunotherapy [8]. Tumor-infiltrating B lymphocytes have been reported in various types of cancers [9] and their presence has been linked to a favorable clinical outcome for some solid tumor types [10]. In addition to the immune-regulatory role of antibody and antibody–antigen complexes, B cells can organize the functions of other immune cells through the presence of antigens, costimulation, and secretion of cytokines that promote immune response to tumors [11,12].

In this review, we will focus on the relationship between EVs and the diverse roles (Figure 1) of immune cells, particularly B cells, in orchestrating multiple aspects of humoral and cellular antitumor immunity, especially as it relates to TAAs. We will also consider current and future utilization of these various components of the immune system for cancer immunotherapy.

## 2. Evidence of the Interaction between TDEs, TAAs, and Tumor Immunity

EVs include various groups of lipid-bound nanoparticles that are 50–1000 nm in diameter and that are secreted by most types of cells in normal and diseased states [13,14]. Consensus has not yet emerged on specific markers or nomenclature for EV subtypes, and EVs are typically classified according to their route of biogenesis or by their physical characteristics that are exploited for separation. Exosomes represent a specific subclass of small extracellular vesicles (sEVs), less than 200 nm in diameter, which emerge by explicit endosomal biogenesis pathways [15]. Medium/large extracellular vesicles (m/lEVs) represent other subsets of EVs and are typically attributed as microvesicles, oncosomes, and apoptotic bodies derived from the plasma membrane [16,17]. EVs are comprised of phospholipids as well as membrane microdomains, such as sphingolipid-enriched lipid rafts and caveolae, and diverse molecular cargoes including nucleic acids, metabolites, and proteins [18]. These vesicle entities are regularly classified and studied and are currently of great interest in research [19,20]. As a caveat, we note that the diversity of EVs secreted by cells along with difficulties in assigning particular biogenesis pathways has led to often indefinite usage of the term “exosome” to refer generally to small EVs, or any EVs recovered after 100,000× *g* ultracentrifugation, for example. In this review, we use the term EV to include all the various lipid bound particles described above. As the field continues to develop, standardized nomenclature and better mechanistic insights will allow for more defined assignment of EV subtypes with specific biological functions.

TDEs are found in abundance in plasma and malignant effusions [21]. TDEs have potential to yield biomarkers for cancer interception, tumor molecular subtyping and disease monitoring [22]. EVs also display tumor-associated antigens and transfer native tumor-derived proteins and antigens to antigen-presenting cells (APCs). TDEs containing native tumor antigens can be efficiently taken up by dendritic cells (DCs) and the antigens processed and cross-presented to naïve T cells [23]. The presence of APCs and expression of TAAs such as prenatal exposed antigens have been found to contribute to suppression of T cell activation and tumor progression [24]. As an antigen-independent T cell reaction, immune checkpoint signaling by exosomal programmed death-ligand 1 (PD-L1) expression has attracted interest. PD-L1 was originally discovered to play a tumor supportive role. When expressed on the tumor cell surface, PD-L1 facilitates evasion of immune surveillance by interacting with programmed death-1 (PD-1), thereby inhibiting T cell function. Metastatic melanomas release EVs that carry PD-L1 and suppress the cytotoxic function of CD8+ T cells [25]. This important EV-mediated mechanism of T cell immune escape has become well established. However, humoral immunity elicits anticancer effects that augment and extend beyond T cells, and there are other mechanisms of EV contribution to antitumor immunity or immune escape that merit additional investigation.

B cells have been shown to be critical mediators of anticancer immunity that extend beyond antibody production to include antigen presentation and activation and modulation of T cells and innate immune effectors. The tumor microenvironment contains a heterogeneous population of B cells, with both protumorigenic and antitumorigenic activity [26]. In high-grade serous ovarian cancer, CD20+ tumor-infiltrating lymphocytes (TIL) were identified as colocalized with CD8+ T cells. Notably, B cell infiltration correlated with increased patient survival compared to the occurrence of CD8+ TIL alone [27]. In another study, gene-based signatures of tumor-infiltrating B cells were found to be predictive of response to immune checkpoint therapy. Specifically, mass cytometry revealed memory B cells to be enriched in the tumor of responders [28]. In another study progression of castration-resistant prostate cancer was associated with B cell infiltration and activation of IKKα, which stimulates metastasis by an NF-κB-independent mechanism [29]. These data suggest spatiotemporal and context-dependent aspects of tumor and B cell interactions have yet to be fully understood.

## 3. EVs and Crosstalk with the Immune System

EVs are versatile effectors of cell–cell communication that mediate multilateral tumor–immune interaction and exchange. Immunological activity of EVs was first reported by Raposo and colleagues with the finding that B cells release MHC class II (MHC-II) antigen-presenting EVs with demonstrated capacity to elicit antigen-specific CD4+ T cell responses [30]. With downstream implications for both cellular and humoral immunity, classical antigen presentation of CD4+ T cells by MHC-II molecules modulate the initiation and progression of the immune activation cascade: activated CD4+ T cells proliferate and differentiate into cytokine-secreting effector T cells that subsequently promote antigen-primed B cells to proliferate and induce class-switch recombination and somatic hypermutation [31]. A significant proportion of MHC-II-bound antigenic peptides are secreted by activated B cells, and engagement of activated B cells with antigen-specific CD4+ T cells further stimulates EVs release from antigen-loaded B cells [32]. Signaling for EV release from B cells can also be elicited by simple MHC-II crosslinking. B cell synthesis of EVs is also initiated following the receipt of various cytokine activation signals, which include interleukins, interferons, and tumor necrosis factors [33,34].

EVs from T cells or DCs can stimulate the proliferation and differentiation of B cells. A series of self-tolerance mechanisms hold autoreactive B cells that emerge in the bone marrow under control. B cells that are explicit in lower-valence autoantigens can reach the peripheral circulation; however, chronic autoantigen exposure prompts IgM downmodulation and diminished BCR binding to downstream pathways, a condition called B cell anergy [35]. Previous studies have demonstrated that viability of autoantigen-engaged B cells is greatly diminished in the absence of CD40L-expressing T cells [36]. With regard to EVs, it is found that EVs released from activated CD4+ T cells play important roles in the activation, proliferation, and differentiation of B cells and CD40L is involved in EVs derived from CD4+ T cells [37]. EVs from DCs have been suggested to bear CD54, CD86, and MHC class I and II and to be able to induce antigen-specific regression of tumor in a murine cancer model [38]. This effect is related to immune cells reaction against DC derived EVs. In mice, elevated Th1 immune responses are induced by DC derived EVs, where the T cell response was dependent on B cells [39]. Moreover, EVs induced increased germinal center B cell proportions [40]. These results are important for elucidating the function of EVs in vivo, and for designing immunotherapies and vaccines based on vesicles.

EVs released from cells of the innate immune system, the first-line protection against infection and cancer, which includes mast cells, neutrophils, macrophages, eosinophils, basophils, and natural killer cells, may also stimulate cell activity. Various types of innate cells can mutually secrete and take up EVs and interact with adaptive immune cells including B cells [41]. For example, mast cells produce EVs whose content is quite heterogeneous. When the anti-CD40L blocking Ab was applied to mast cell derived condition medium and grown with B cells, a substantial decrease in proportion of CD19+IL-10+ B cells was observed relative to B cells cultured in mast cell condition medium. In addition, mast-cell-derived CD40+ EVs influenced isotype switching and plasma and memory B cell formation [42]. Mast cells are known to generate EVs containing GTP-activatable phospholipases and bioactive lipid mediators, including PGD2 and PGE2, which can exert immunomodulatory effect [43]. Studies on hematopoietic PGD2 synthetase (PTGDS) KO mice showed decreased IL-10 and increased TNFα secretion from B cells compared to that in WT control mice [44]. Other study demonstrated PGE2 suppressed B cell proliferation by using B cell lymphoma [45,46]. Together, these data suggest prostaglandins conveyed by EVs could play an essential role in attenuating B cell activation.

Macrophages are primary responders of innate immunity that play crucial roles in antigen presentation, phagocytosis, and immunomodulation [47]. Induction of M1-like phenotype macrophages prompts immunostimulation, with an expansion of proinflammatory cytokines and chemokines, coordinated at riddance of pathogenesis and infection [48]. Breast and gastric TDEs are able to activate a proinflammatory response in M1-like macrophages through the induction of NF-κB signaling that initiates transcription of inflammatory cytokines including GCSF, IL-6, IL-8, IL-1β, CCL2, and TNF-α [49]. Cytokine IL-1β lacks a canonical signal peptide, and in macrophages, EV-mediated release is a major secretory pathway for IL-1β, which in turn modulates function and migration of DCs as well as the expansion and differentiation of T and B cells [50,51].

## 4. The Effect of B-Cell-Derived EVs against Other Immune Cells

Exosomes from B cells have been found to employ both MHC class I and II proteins on their surface for antigen presentation, and a number of groups have expanded on this to demonstrate an antitumor effect for immune cell exosomes [52]. In this context, EVs secreted by both human and murine B cells, which express MHC molecules on their surface, induced antigen-specific MHC-II-restricted T cell responses [30]. It has also been shown that B cells and the antibodies they produce play crucial roles in the activating T cell response to protein antigen [53]. In vivo studies have recently suggested that exosomes are more effective in the introduction of intact protein antigen than peptide [54]. Host, instead of exosomal MHC-I, was found to be required to present an antigen to CD8+ T cells, regardless of the significant expression of MHC-I on primary B cell EVs [55]. Induction of cytotoxic T lymphocyte (CTL) response by B-cell-derived EVs was found to be highly reliant on the presence of CD4+ T cells, CD8+ T cells, and NK cells. Absence of any of these immune cell subtypes (CD4+, CD8+ T-cells, NK cells) resulted in a complete loss of B-cell-derived EV capacity to mediate CTL response. Interestingly, host B cells, the BCR, and B-cell-secreted antibodies were further shown to not play a direct role in facilitating the CTL response initiated by exogenous B-cell-derived EVs, based on in vivo studies of B cell depletion or utilization of a mouse model with B cells that lack membrane-bound and secreted antibody [56].

In many cases, B-cell-derived EVs provide proimmune stimulus. However, CD19+ EVs from B cells, which include high CD39 and CD73, are reported to deplete chemotherapeutic effectiveness by inhibiting CD8+ T cell responses. CD39 is an ectonucleotidase that hydrolyzes proinflammatory ATP and ADP to AMP; whereupon, CD73 catalyzes conversion of AMP to adenosine. Adenosine exerts immunoregulatory functions through direct binding to one of four adenosine receptors: A1R, A2AR, A2BR, and A3R. Subclassification of adenosine receptors is largely based on receptor ability to induce the downstream signaling molecule cyclic AMP (cAMP); cAMP signaling through A2AR and A2BR is predominately associated with potent immunosuppression, whereas activation of A1R and A3R attenuates cAMP biogenesis and is, therefore, typically considered to be immune-promoting [57,58,59,60]. There is evidence that these receptors are differentially expressed on various immune cell types. A2AR is primarily associated with T cells where cAMP elevation results in suppression of TCR function and IFN-γ production [61]. Chemotherapy induces the copious release of ATP from tumor cells. CD39 and CD73 expressed on EVs appears to convert this ATP to adenosine within the tumor milieu, thereby attenuating antitumor T cell responses. Inhibiting Rab27a via inactivated Epstein–Barr virus-mediated transfer of siRNA depleted the production of CD19+ EV in B cells and considerably improved the chemotherapy effect [62,63,64]. The other way which B-cell-derived EVs control immune system negatively is directed by FasL upregulation [65]. FasL presentation evoked apoptosis of antigen-specific CD4+ T cells, indicating a negative reaction to control T cell responses [66]. EVs obtained from cultured primary B cells and infused into mice were found to be detected by macrophages in the subcapsular sinus of lymph nodes. This mechanism includes binding of exosome- linked α2,3-linked sialic acids to CD169 in lymph nodes or spleen [54]. From these evidences, B cell EVs derived outside the lymphoid tissue are found to be eliminated by macrophages, likely as a method for modulating immune responses.

One of the targets of B-cell-derived EVs is follicular DCs. Although follicular DCs do not endogenously synthesize MHC-II, follicular DCs are decorated with MHC-II-carrying B cell EVs at their plasma membrane [67]. The idea that these EVs might be gained considerably from B cells is developed by the perception that isolated B cell exosomes are effectively recruited from tonsil lymphoid tissue by follicular DCs, when contrasted with other cell types within the same environment [68]. All things considered, follicular DCs, using FcR, enroll B cell EVs and complement receptors. Such binding may occur with integrin α4β1, which is also highly abundant and functional on EVs derived from B cells [69]. B cell EVs convey complement 3 (C3) fragments and BCR–antigen complexes [70], and it appears that C3 fragments on these EVs induce a T cell response in the presence of suboptimal antigen concentrations [71].

## 5. Mechanism of EVs Action for B-Cell-Derived Humoral Immunity

In systemic autoimmune disease, autoantibodies produced by B cells react with free molecules, cell surface, and nucleoprotein antigens, as well as complex glycosphingolipids, forming antigen–antibody complexes [72]. TDE expression of tumor antigens may serve to attenuate binding of antibody to tumor antigens, suggesting a role for tumor-secreted EVs in tumor evasion from the effect of autoantibodies [73]. In plasma of ovarian cancer patients, six common immune-reactive proteins, including GRP78, annexin 2, cathepsin D, alpha-enolase, HSC70, and PDI, were identified by mass spectrometry. The identity of autoantibodies against these proteins was also confirmed by immune-reactivity with recombinant proteins. EVs were isolated from ascites fluid and bound-immunoglobulins (Igs) eluted. The EV-associated Igs exhibited recognition of recombinant annexin 2, HSC70, and GRP78 [73].

Capello et al. recently conducted experiments to specifically explore the relationship between EVs and autoantibodies [6]. To this end, in-depth proteomic profiling of immune complexes was performed using plasma samples of patients with PDAC. Localization analysis of gene ontology revealed statistically significant enrichment of EV/exosome- and endosome-associated proteins among the Ig-bound proteins. This finding revealed the possibility that intact EVs from PDAC patients bind to circulating Ig molecules. Affinity-purified plasma Ig fractions from PDAC patients were subjected to EV isolation using density gradient ultracentrifugation (UC). Nanoparticle-tracking analysis and transmission electron microscopy indicated the presence of sEVs in the UC sample from the Ig-bound fraction. For further confirmation, flow cytometry analysis was performed, which revealed high levels of human IgG in the isolated exosomes from PDAC patients’ plasma. Proteomic profiling of PDAC cell line EVs was conducted as well. The presence of TAAs associated with autoantibodies was investigated using the EV surfaceome from a PDAC cell line, which resulted in significant enrichment of TAAs based on Cancer Immunome Database analysis. Comparing the Ig-bound fraction of PDAC patient plasma to cell line EVs, 34 of the 92 Ig-bound proteins were identified in the PDAC exosome surfaceome.

Presentation of TAAs on MHC (human, HLA) molecules is necessary to induce T-lymphocyte activation and autoantibody production [74]. To assess whether plasma EVs in PDAC patient plasma include HLA molecules, EVs derived from the Ig-bound fraction of PDAC patient plasmas were compared with HLA-associated peptides from PDAC cell lines. Peptides from ten proteins (ACTB, GSTP1, HSPA8, KRT10, KRT20, KRT5, PCBP1, PKM/PKM2, TUBB, and UBA52) were concordant between PDAC EVs from patient plasma and PDAC cell lines. With the promotion of complement-dependent cytotoxicity (CDC), autoantibodies can mediate cytotoxic effects on cancer cells following surface TAA ligation. A pool of prediagnostic PDAC patient serum samples induced CDC of cells from PANC-1 cell line. When EVs from the same cell line were added to the culture, CDC was significantly inhibited in dose-dependent manner. This finding demonstrates that cancer-associated EVs can effect a decoy function that modulates CDC response by competing for autoantibody binding.

The biological role and clinical importance of the tolerogenic leukocyte antigen, HLA-G have been intensively investigated. HLA-G has seven isoforms together with four membrane-bound (HLA-G1, -G2, -G3, and -G4) and three soluble (HLA-G5, -G6, and -G7) isoforms [75]. Some of the known functions of HLA-G is limited to the surface-expressed receptors of the innate and adaptive immune cells. Thus, HLA-G inhibits the proliferation, differentiation, and immunoglobulin secretion of B cells [76], the cytolytic function of NK cells [77], the antigen-specific cytolytic function of CTL [78], proliferation of CD4+ T cells [79], and the maturation and function of DC [80]. HLA-G-bearing EVs are secreted from cytotrophoblast cells, mesenchymal stem cells, and cancer cells [81]. TDEs with HLA-G can require the interaction with target cells that lack the surface expression of HLA-G specific receptors to effect cancer immune escape [81].

An intriguing function for TDEs is in relation to the antigen presentation mechanism referred to as MHC cross-dressing. There is evidence that MHC molecules loaded with antigenic tumor peptides can be transported by tumor-derived EVs to professional APCs like B cells [21,82]. Host DCs cross-dressed with tumor MHC molecules will show new TAA epitopes to CD4+ and CD8+ T cells that are not endogenously formed by the presenting APCs [83]. MHC cross-dressing of APCs provides an antigen presentation pathway for the T cells by acquiring preformed peptide–MHC complexes, which are alternatives to control and cross-presentation. Apparently, the T-cell-stimulating potential of APC-derived EVs is that, even for naïve T cells, particularly when the EVs are introduced bound to the surface of DCs. On the EV surface, ligands including lactadherin, tetraspanins, externalized phosphatidylserine, C-type lectins, and CD54 bind by bridging molecules to receptors on target APCs [84]. Exogenous EVs bind to DCs, some of which remain on the surface, and therefore internalize the remnants for antigen processing and regular cross-presentation. A few factors regulate the potency of MHC cross-dressing of APCs via EVs, and acceptor APCs behave as platforms that intensify the capacity of the EVs generated to express peptide-MHCs straight to T cells. This EV-mediated move of preformed MHC–peptide complexes to APCs is being developed as a cancer vaccination strategy [85].

## 6. Opportunities for Intervention Utilizing EVs and B Cell Immune Response

B cells mediate humoral antitumor responses, including complement-dependent cytotoxicity, antibody-dependent cellular cytotoxicity (ADCC), and antibody-dependent cell phagocytosis (ADCP), with implications for therapeutic benefit. B cells also function in cellular immunity through antigen cross-presentation, eliciting stimulation, and secreting cytokines that cooperate to ultimately effect T cell activation [26]. TDEs can disseminate tumor antigens and prompt antitumor immunity. However, TDEs can also exert decoy function against antibody-mediated antitumor processes and promote tumor resistance to host responses or therapeutic intervention. In this context, TDE control can be harnessed for specific cancer therapy. Early studies of sEV/exosome release identified ceramide to be involved in the inward budding of the endosome membrane that yields multivesicular bodies (MVBs) containing exosomes; ceramide generation is catalyzed by the enzyme neutral sphingomyelinase (nSMase2). Exosomes and sEV populations are enriched in ceramide. Their secretion is known to be reduced by inhibiting SMase2 with siRNA or the small molecule inhibitor GW4869 [86]. GW4869 treatment for B16BL6, a mice melanoma cell line, significantly decreased tumor growth in both in vitro and in vivo models [87]. Other similar lipid metabolism inhibitors, Pantethine and Imipramine, also inhibit production of EVs [88,89]. On the other hand, Calpeptin is a compound that controls EV trafficking by inhibiting calpain. Calpain is a Ca^2+^-dependent protease of the papaya proteinase family of the cysteine proteinase superfamily, which is spread widely in the cytoplasm of most mammalian tissues; specifically, the plasma membrane and the membrane-bound organelles are associated [90]. In a prostate cancer cell line model, inhibition of EV release by Calpeptin resulted in accumulation of docetaxel and methotrexate treatment within cells, resulting in significantly reduced cell proliferation and increased cytotoxicity compared to that observed without of calpeptin treatment [91]. Manumycin A is another EV trafficking inhibitor that suppresses Ras. Ras is small GTPase able to regulate the function of other proteins to tumorigenic effect, including proteins involved cellular processes such as cell differentiation, cytoskeletal integrity, cell adhesion and migration, cell proliferation, EV release, and apoptosis. Manumycin A decreases the viability of cancer cells and EV-associated mechanisms by inducing nonapoptotic, nonautophagic cytoplasmic vacuolation death in triple negative breast cancer cells [92].

An additional approach to attenuate circulating EVs has been described that employs extracorporeal hemofiltration using affinity plasmapheresis. It has been proposed as a means of overcoming the risks of toxicity and drug interactions posed by pharmacological approaches [93]. Continuous whole blood ultrapheresis was used to remove low molecular weight proteins, including EVs, with the result that 6 out of 16 cancer patients had a minimal 50% reduction in tumor size [94]. An alternative, targeted approach would be to selectively deplete tumor-associated EVs from the circulation using specific surface markers to affinity-capture tumor secreted vesicles [6,95]. Recently, nanosponges and nanokillers (NSKs) were developed as new material for capturing TDEs. They are assembled from platelet and neutrophil hybrid cell membrane structured on gold nanocages. In breast cancer cells, NSKs showed higher cell uptake, deeper tumor penetration, and higher cytotoxicity to tumor cells compared to noncoated control nanocage both in vitro and in vivo [96].

The decoy function of EVs can affect not only host autoantibodies, but also administered antibody therapeutics, thus impairing drug efficacy. Antibody based immunotherapy against CD20 is a standard regimen for malignant lymphoma. However, TDEs from lymphoma that carry CD20 shield target cells from antibody binding. The lysosome-related organelle-associated ATP-binding cassette transporter A3 (ABCA3) mediates the biogenesis of these tumor-derived EVs. Both the pharmacological blockade and the silencing of ABCA3 have been shown to increase the sensitivity of target cells to antibody-mediated lysis [97]. EVs derived from HER2-positive breast cancer cells also exert a drug-resistant function for Trastuzumab. EVs released by HER2-overexpressing tumor cell lines expressed a full-length HER2 molecule. Functional assays showed that both xenogeneic and autologous HER-2-positive EVs, but not HER2-negative ones, inhibited Trastuzumab involvement in SKBR3 cell proliferation [98].

The search for new TAAs has started recently. Multiple therapeutic cancer vaccines designed to stimulate helper T cells have been developed by these discoveries [99]. At this stage, several clinical trials involving DC-based vaccines have been conducted to treat tumors of various origins. Nonetheless, there are a number of downsides to DC-based vaccines, as well as concerns about targeted delivery of tumor antigens to DCs and extended cellular vaccines storage [100]. EVs have emerged as alternative, cell-free antitumor vaccines as EV-based vaccines can be stored for an longer period of time and EVs have been shown to be more effective in capturing compared to the soluble molecules in antigen-presenting cells [101]. EVs are therefore more biocompatible and biodegradable, and thus have low toxicity and immunogenicity [102]. One of the effective vaccines which has antitumor potential is tumor-cell-derived DC-targeted EVs. DCs pulsed by K562 leukemia cell line derived EVs (LEX), only LEX and nonpulsed DCs were compared by CTL immune response in vivo and LEX pulsed DC had significant higher response than the other two groups [103]. However, the preparation of TDE-based antitumor vaccines should be combined with comprehensive study of their complete biological effects on the immune system, since TDEs are known in certain contexts to exert immunosuppressive effects [100]. Another effective method is DC-derived EVs. These DC-secreted EVs will convey all the molecules required to cause an immune response in T cell. HPV early antigen 7 peptides (E7_49–57_), which is the primary target antigen in cervical malignancy, was utilized to load murine EV-producing DCs. Treatment effect of E7_49–57_-loaded EVs for cervical cancer-bearing mice instigated an antitumor CTL response in vivo [104].

## 7. Conclusions

There is clear evidence of involvement of EVs in modulating tumor–host immune interaction and tumor responses to immunotherapy. EVs originating from cancer, immune, and other cell types effect immune suppressive as well as immune activating functions. Various subtypes of EVs are known to be involved in antigen presentation, immune signaling, and immune cell crosstalk and communication. In this review, we have surveyed current knowledge regarding these EV functions in the context of cancer, especially in relation to B-cell-mediated activities. Tumor-derived EVs can support tumor progression through acting upon immune system effectors including B cells, T cells, NK cells, and DCs. Immune cell derived EVs coordinate immune system antitumor responses by mediating pan-immune communications between B cells, CD4+ and CD8+ T cells, DCs, mast cells, and macrophages. As summarized in Table 1, EVs play important roles within both the tumor microenvironment and systemic contexts. As a result, there is an opportunity to manipulate EV involvement through antitumor therapy. In this rapidly evolving field, we have highlighted important aspects of these emerging mechanisms and look forward to further development of knowledge in the area that can be applied to future-generation antitumor immune immunotherapies.

## Figures and Tables

**Figure 1 cells-09-01518-f001:**
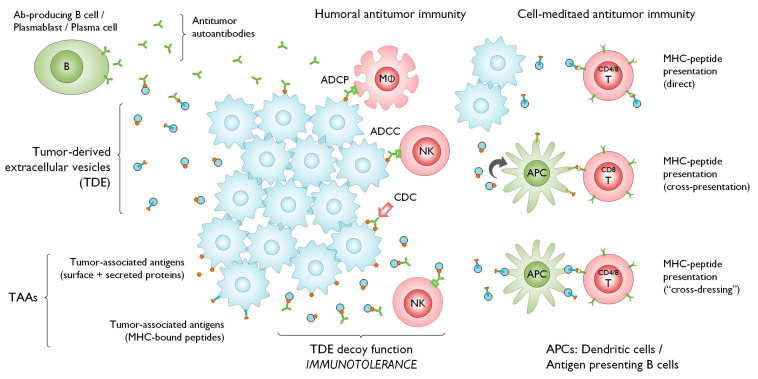
Tumor-derived extracellular vesicles (TDEs) dynamically mediate tumor B cell immune interaction. B cells are known to play roles in humoral as well as cell-mediated antitumor immunity. TDEs convey antigenic proteins and tumor-processed antigenic-peptide-loaded MHCs that complex with circulating plasma anticancer autoantibodies produced by various B cell subpopulations. TDEs function as competitive inhibitor decoys that bind antitumor antibodies and attenuate antibody-dependent cytotoxic and phagocytic immune functions. TDEs can activate CD4+ and CD8+ T cells by presenting MHC-bound antigenic peptides via direct and indirect presentation. B cells can function in the tumor microenvironment as antigen-presenting cells that process TDE antigenic protein cargos for cross-presentation to CD8+ T cells. TDEs expressing MHCs loaded with tumor-peptides can be surface presented directly by B cell without further processing (“cross-dressing”). These mechanisms have utility for developing tumor antigen targets, antitumor vaccines, and suggest strategies for mitigating EV-mediated immune escape to support next-generation cancer immunotherapies. ADCC, antibody-dependent cytotoxicity; ADCP, antibody-dependent cellular phagocytosis; APC, antigen-presenting cell; CDC, complement-dependent cytotoxicity; MΦ, macrophage; NK, natural killer cell.

**Table 1 cells-09-01518-t001:** Tumor- and immune-cell-derived extracellular vesicles (EVs) mediate multilateral networks of cross-communication.

EV Origin	Target	Interaction	Key Molecules	Refs
TDEs	DCs	cross-presentation to naïve T cells and suppression	TAA	[23,24]
TDEs	CD8+ T cells	CD8+ T cell suppression	PD-1, PD-L1	[25]
TDEs	B cells	tumor evasion from antibody-mediated immune response	GPR78, annexin 2, cathepsin D, alpha-enolase, HSC70, PDI	[73]
TDEs	T cells, B cells	T cell activation and autoantibody production	MHC-II molecules (ACTB, GSTP1, HSPA8, KRT10, KRT20, KRT5, PCBP1, PKM/PKM2, TUBB, and UBA52 in PDAC)	[6]
TDEs	B cells	decoy function for CDC and ADCC	TAA	[6]
TDEs	NK cells, CTLs, CD4+ T cells, DCs, B cells	inhibition of NK cell, CTL, CD4+ T cell, and DC	HLA-G	[76,77,78,79,80,81]
TDEs	DC, B cell	MHC cross-dressing	MHC molecules, lactadherin, tetraspanins, externalized phosphatidylserine, C-type lectins, CD54	[21,82,83,84]
TDEs	CD20 targeting antibody	TDE binds to CD20-targeting antibody and inhibits the effect	CD20, ABCA3	[97]
TDEs	Trastuzumab	HER-2 positive EVs inhibit Trastuzumab	HER-2	[98]
TDEs	DCs	cancer vaccine (activate B cells)	LEX, TAA	[103]
B cell EVs	CD4+ T cells	antigen presenting	MHC-II	[30]
B cell EVs	CTLs	CTL activation	MHC-I, antigen	[55,56]
B cell EVs	CD8+ T cells	inhibition of CD8+ T cells and attenuation of chemotherapy efficacy	CD39, CD73, Adenosine	[62,63,64]
B cell EVs	CD4+ T cells	induction of apoptosis for CD4+ T cell	FasL	[65,66]
B cell EVs	macrophages	attenuate immune response outside the lymphoid tissue	α2,3-linked sialic acids, CD169	[54]
B cell EVs	follicular DCs	decorate with MHC-II carrying EVs	MHC-II, FcR, integrin α4β1	[67,68,69]
B cell EVs	T cells	stimulate T cell response	C3, BCR	[70,71]
CD4+ T cell EVs	B cells	differentiation of B cell	CD40	[36,37]
DC EVs	B cells	increase of germinal center B cell	CD54, CD86, MHC-I and II	[38,39,40]
DC EVs	T cells	cancer vaccine (activate B cell)	TAA (HPV antigen E7_49-57_)	[104]
Mast cell EVs	B cells	CD19^+^IL-10^+^ B cell increase	CD40L	[42]
Mast cell EVs	B cells	B cell attenuation	PGD2, PGE2	[43,44,45,46]
MΦ EVs	DCs, T cells, B cells	migration of DCs, expansion and differentiation of T and B cells	NF-κB, IL-1β	[49,50,51]

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
