# Peer review of "Extracellular Vesicles Mediate B Cell Immune Response and Are a Potential Target for Cancer Therapy"

_cells, 2020, doi:10.3390/cells9061518_

Round 1
Reviewer 1 Report
This work by Kato T et. al, describes the role of EVs and their relationship with the immune system, with greater emphasis on the role played by vesicles in the function of B cells. Finally, providing potential clinical utility of these molecules on immunotherapy.
some minor comments,
Line 19. Evidence of B cell autoimmune response against tumor-associated antigens (TAAs) is observed in early stages of tumorigenesis an in most solid tumor types.
Suggest just mention, Evidence of B cell response against tumor-associated antigens (TAAs). The word autoimmune is not used to refer to the antitumor response.
Line 22. plasma anti-cancer autoantibodies. Suggest; replace by antitumoral antibodies
Line 69. The presence of these immune cells and expression of TAAs have been found to contribute to suppression of T cell activation and tumor progression.
Explain which immune cells , DCs??? it refers to and the mechanism by which it performs this function
Line 92. Immune crosstalk and induction of B cell derived EVs. Suggest; EVs and crosstalk with immune system
Line 155. B cell-derived exosomal immunity demonstrated a definite dependence on CD4+ T cells, CD8+ T cells and NK cells, with the decrease of each subset resulting in a whole loss of immune response in cytotoxic T lymphocyte (CTL). Please, explain better this sentence, it is not understood what it refers specifically
Line 163. Chemotherapy induces the copious release of ATP from tumor cells, which are then hydrolyzed into adenosine by CD39 and CD73 and decrease the response of antitumor T cell responses.
Please, Please explain better the sentence and the mechanisms associated with the decrease in the antitumor response.
Line 263. B cells produce autoantibodies. The B cells produces more than autoantibodies, in some circumstances they are produced but this sentence highlights that this is the role of these cells. I suggest removing auto antibodies
Reviewer 2 Report
This is a well written and comprehensive review about the roles of extracellular vesicles (EVs) and B cells in tumorigenesis and about their potential therapeutic application. I suggest some minor changes:
- Line 55-62: according to the ISEV (International Society for Extracellular Vesicles) 2018 guidelines, EVs can be clustered either based on their cellular origin (exosomes, microvesicles, apoptotic bodies) or separation features (small and large EVs). It would be good to strictly follow these recommendations and to insert a short paragraph about general problems with EV clustering. Since older works often used different nomenclatures for EVs, I would suggest to discuss briefly that "exosomes" in these papers do not necessarily mean EVs of endosomal origin, but more general EVs.
- Conclusions, 334-340: I would suggest to describe briefly the content of this review again with mentioning the major discussed points.
